# Delay-Packet-Loss-Optimized Distributed Routing Using Spiking Neural Network in Delay-Tolerant Networking

**DOI:** 10.3390/s23010310

**Published:** 2022-12-28

**Authors:** Gandhimathi Velusamy, Ricardo Lent

**Affiliations:** College of Technology, University of Houston, Houston, TX 77204, USA

**Keywords:** delay-tolerant networking, spiking neural network, satellite communication, QoS, AI, ISL

## Abstract

Satellite communication is inevitable due to the Internet of Everything and the exponential increase in the usage of smart devices. Satellites have been used in many applications to make human life safe, secure, sophisticated, and more productive. The applications that benefit from satellite communication are Earth observation (EO), military missions, disaster management, and 5G/6G integration, to name a few. These applications rely on the timely and accurate delivery of space data to ground stations. However, the channels between satellites and ground stations suffer attenuation caused by uncertain weather conditions and long delays due to line-of-sight constraints, congestion, and physical distance. Though inter-satellite links (ISLs) and inter-orbital links (IOLs) create multiple paths between satellite nodes, both ISLs and IOLs have the same issues. Some essential applications, such as EO, depend on time-sensitive and error-free data delivery, which needs better throughput connections. It is challenging to route space data to ground stations with better QoS by leveraging the ISLs and IOLs. Routing approaches that use the shortest path to optimize latency may cause packet losses and reduced throughput based on the channel conditions, while routing methods that try to avoid packet losses may end up delivering data with long delays. Existing routing algorithms that use multi-optimization goals tend to use priority-based optimization to optimize either of the metrics. However, critical satellite missions that depend on high-throughput and low-latency data delivery need routing approaches that optimize both metrics concurrently. We used a modified version of Kleinrock’s power metric to reduce delay and packet losses and verified it with experimental evaluations. We used a cognitive space routing approach, which uses a reinforcement-learning-based spiking neural network to implement routing strategies in NASA’s High Rate Delay Tolerant Networking (HDTN) project.

## 1. Introduction

Satellite communication has been used as: 1. Satellite and terrestrial integrated networks (STINs) to extend terrestrial network infrastructure for providing global broadband communication; 2. Space-based information networks—satellite networks used to collect Earth observation data and planetary exploration mission data.

### 1.1. Satellite and Terrestrial Integrated Networks

The network traffic has increased tremendously in recent years because of everything connected to the Internet at anytime (IoT) [1] and the increased use of remote services after the COVID-19 pandemic, which required the satellite network to be integrated with the terrestrial network to fulfill the network demands [2]. Hence, satellites are used in the communication industry by providing:Service continuity in areas where terrestrial networks are not availableService ubiquity to provide resilient service during terrestrial network failuresScalability to load-balance the traffic demands exceeding the terrestrial network’s capacity [3].

### 1.2. Space-Based Information Networks

Space-based information networks consist of satellite constellations at different Earth orbits, including: enumerate 1. Earth Sensor Web and other satellites used for investigating planets such as Mars and the Moon in the solar system; 2. Network infrastructure that connects satellites to the backbone, access, and proximity networks [4]. enumerate Earth observation and monitoring systems collect time-sensitive information about the Earth using remote sensing satellites [5,6]. Constellations of a large number of small satellites are used to collect simultaneous and distributed measurements or observations in monitoring Earth resources, weather, and disaster situations with an increased temporal resolution of collected data [7]. Besides, the National Aeronautics and Space Administration (NASA) is ambitiously developing future space and ground architecture for deep space and planet exploration to meet the final objectives of lunar and Mars exploration, habitation, and colonization [8,9]. The future space architecture will provide communications, navigation, and inter-networking services for space missions from Earth’s orbit to Mars and other planetary exploration missions.

### 1.3. Challenges in Space Communication

The satellite network maintains three types of links [10]. Each kind of link has issues that affect the performance of the satellite networks:Inter-satellite links (ISLs): links between satellites in the same layer; for example, satellites in LEO have four ISLs to connect with four neighbors on the same orbit. Satellites in MEO connect with their immediate neighbors in their orbit.Inter-orbital links (IOLs): satellites in different orbits communicate through IO; for example, communication links between GEO and MEO, GEO and LEO, and MEO and LEO.User data links (UDLs): communication links between satellites and ground stations, also known as feeder links. A satellite can maintain several UDLs to multiple ground stations, and a ground station can directly connect to many satellites in any orbit.

The data transmission from satellites at the lower Earth orbits suffer from the short contact windows to ground stations in each pass and are subject to highly dynamic topological changes due to the high speed of satellites. High-frequency free space optics channels in the feeder links between satellites and ground stations help to deliver data with high throughput. However, free-space optics channels are susceptible to attenuation due to unpredictable weather conditions [11]. Site diversity in ground stations helps alleviate problems due to short contact windows, but installing and maintaining multiple ground stations at several geographical locations involve huge investments [6,12,13].

The introduction of ISLs provides a promising solution to solve packet loss rates and long delays due to atmospheric attenuation and the short contact windows associated with direct satellite-to-ground station communication by utilizing satellite hand-overs. The higher-bandwidth ISLs between satellites on the same orbits and other orbits established by laser and radio terminals help to transmit data between satellites at high rates [6]. However, ISLs suffer from dynamic topological changes due to the inter-plane ISLs between satellites in different orbits shutting down and reestablishing (on–off switching) in and out of polar regions [14,15]. Besides, satellites in the 0th and (N − 1)th planes across the seam cause the absence of ISLs and need to detour to access satellites on the other sides, which causes longer delays [16]. Another issue with ISLs is that the traffic load on the ISLs varies according to the demographic distribution of the coverage areas on the Earth [17]. The non-uniform traffic load on the celestial network is due to population dispersion, economic status, and technology penetration status over different geographical areas [18].

The IOLs have line-of-sight issues due to the movement of satellites in their orbits and suffer longer delays due to the distances between various orbits [19]. When multiple satellites try to relay data through a single satellite in a different layer to use the shortest path, this causes congestion and leads to packet losses [19].

Hence, it is highly challenging to design routing strategies adaptable to the link dynamics of ISLs, IOLs, and UDLs in delivering data from satellites to ground stations with an optimized delay and packet loss rate to fulfill the performance requirements of the applications they serve.

The existing works on satellite networks with multi-QoS optimization routing goals use a linear combination of the metrics such as the delay, packet loss rate, and throughput with weights assigned to each. They try to optimize any one metric according to the requirement of the application by giving more importance through the weight. Hence, the goal of simultaneously minimizing both the delay and packet loss will be a challenging problem, for which we have yet to find a solution. Besides, the routing approaches that optimize delay may use paths that give short delays due to the loss of packets and suffer reduced throughput. This premise has been theoretically proven and validated using simulation results by considering packet losses due to the available buffer capacity in DTN nodes [20].

### 1.4. Our Routing Approach

This work investigates packet losses due to the channel characteristics while all satellite nodes have equal buffer capacity. We tried to optimize latency and packet loss together without any weights or priorities by using the simplified Kleinrock’s power metric [21] in the snnr routing algorithm, which uses the ratio of delay and packet loss. We compared the performance of snnr with the other routing approaches with the following goal functions:1. Linear combination of delay and packet loss; 2. Delay-only; 3. Packet-loss-only; 4. Contact graph routing (CGR), which uses the shortest path with the earliest available links. We evaluated the performance of the routing approaches in a laboratory testbed using virtual machines under various testing scenarios with different emulated delay, packet loss, and bandwidth configurations. We implemented our routing algorithms using the CSG routing approach with the HDTN software running on each virtual machine. Though we briefly explain HDTN and CSG in the following subsections, the readers are encouraged to read the references for complete insights.

### 1.5. HDTN

With the advancements in optical communication technology used in space communication such as with the Laser Communications Relay Demonstration mission, it is possible to achieve data rates up to 1.244 Gbps [22]. However, the existing DTN implementations, interplanetary overlay networks (IONs), are limited in processing such high rates due to the lack of parallel processing capabilities and shared memory. To overcome the rate asymmetries in space data communication, HDTN was developed by NASA. HDTN uses high-performance computing elements in space gateways, which are size, weight, and power constraint platforms (SWap), to provide DTN services to high-rate space communication links (optical, laser) by optimizing the network processing tasks [23]. Currently, HDTN is used as a DTN gateway for the International Space Station to store and forward scientific payloads. It is used for network flow management using a distributed and service-based approach. HDTN supports optical data rates and cognitive networking capabilities [24].

### 1.6. CSG

The CSG approach was developed to optimize bundle routing in delay-tolerant networking using a machine learning approach by leveraging the cognitive networking support offered by HDTN. A cognitive network controller (CNC) continuously updates the synapse strengths (weight) of the neurons in the spiking neural network using reinforcement learning. The number of vertices of the SNN is V=3n+1, and the number of edges is W=n(n+2), where *n* is the number of outbounds at a satellite node. The SNN has a core neuron for each outbound link. Each core neuron is connected to one excitatory and one inhibitory neuron. All core neurons are connected to an inhibitory neuron using post-synaptic connections [25]. The synapse weights are updated using a reward, which is mapped to the optimization goal function and updated based on the performance of the selected outbound link. The neurons emit spikes when their membrane potential reaches a threshold. The outbound link associated with the neuron that emits a spike at the earliest will be selected. Further details can be found in the literature [25,26]. CSG uses forward and backward reports to collect metrics used for helping the CNC to make dynamic routing decisions.

### 1.7. Advantages of Using SNN for Satellite On-Board Routing

Reinforcement learning (RL) is suitable for highly dynamic space networks since it does not need prior training. The SNN uses temporal encoding of the data as an internal mechanism to learn the relationship between input variables related to spatio-temporal patterns that need to be learned, classified, and predicted. Hence, the SNN can be used for predicting early events such as strokes and heart failures in the medical field and earthquakes in environment monitoring [27]. Therefore, it is appropriate for detecting changes in the channel conditions and dynamic path selection in satellite routing. Since the SNN uses spikes, which are event-driven and sparse in time, SNN algorithms consume less energy than other neural networks [28]. Therefore, the SNN is advantageous to be used in on-board routing in satellites to save battery energy.

## 2. Related Work

A large body of literary works have become available for routing in LEO satellite networks used for 5G/6G integration in recent years. In contrast, our work focuses on layer-agnostic routing in satellite networks used for space data delivery. We classified our related work into multi-layered satellite routing, LEO satellite routing, routing in Earth observation constellations, and multi-objective optimization routing in satellite networks.

### 2.1. Routing in LEO Satellite Networks

A load-balancing routing method based on the extended link states (LRES) algorithm, which uses a congestion avoidance mechanism, has been proposed to select a path from multiple paths between a source and a destination node in satellite constellations [29]. A cooperative data downloading (CDD) routing algorithm for LEO networks that use the inter-satellite laser link between the visible satellites to schedule feeder links’ bandwidth resources has been proposed to optimize remote sensing data downloading [30]. An autonomous on-board routing algorithm that makes routing decisions on each hop using the link state information of the ISLs between neighboring satellites was proposed for LEO satellite networks [31]. The location-assisted on-demand routing protocol (LAOR) computes the shortest path for each communication request to estimate a better network state and avoid congestion in the LEO satellite network [18]. An adaptive routing for the non-geostationary satellites network was proposed for dynamic network topology changes and traffic load fluctuations on the ISLs using a link cost metric comprising propagation delay and traffic load [17]. A clustering routing scheme based on the Nash bargaining solution that contains intra-cluster and inter-cluster routing phases was proposed using an agent-based clustering framework [32]. An extreme-learning-machine-based distributed routing (ELMDR) strategy was proposed to make routing decisions on LEO satellites based on the traffic forecast at the satellite nodes using the ground traffic load [33].

### 2.2. Routing in Multi-Layer Satellite Networks

A multi-layered satellite routing algorithm (MLSR) was proposed for a multi-layered satellite IP network comprised of GEO, MEO, and LEO satellites to calculate routing tables efficiently using delay measurements [10]. A survivable routing protocol was presented to provide the ability to survive under LEO or MEO satellite failures and minimum delay routing using a topology control strategy in LEO/MEO satellite networks (LMSNs) [34]. A QoS-aware load-balancing was proposed to alleviate the delays and improve throughput in LEO satellite networks by congestion-prediction-based detouring of the traffic via MEO satellites [35]. A traffic distribution from the LEO to the MEO layer to minimize packet delivery delay in multi-layered satellite networks was proposed by considering propagation and queuing latency [36]. An ant-colony-based inter-layer link handoff algorithm was proposed to reduce the number of inter-layer link handoffs by considering the inter-link layer distance, duration, and QoS metrics [19].

### 2.3. Routing in Earth Observation Satellite Constellations

A novel routing method was proposed to improve the transfer ratio of the whole observed data in a contact time by dividing the data and using them to modulate multiple carriers to send EOS data into two or more ground stations by multi-hop relaying [37]. A joint space–temporal routing algorithmic framework was devised in which disruption-tolerant-networking-based Earth-observing satellite networks with a frequently changing topology and sparse and intermittent connectivity were modeled as a space–time graph [38]. A minimum-cost constrained multipath (MCMP) algorithm was used to find a feasible set of available paths to send a certain amount of mission data to ground stations within a tolerable delay. In the same work, the earliest arrival multipath routing policy based on the contact graph routing was proposed. A cognitive engine architecture that used neural-network-based reinforcement learning (RLNN) was proposed to configure space links from NASA’s International Space Station testbed to the ground station to achieve multiple optimization objectives such as delay, throughput, bandwidth, and power [9].

### 2.4. Multi-Objective Optimization Routing in Satellite Networks

Heuristic-based QoS-oriented satellite routing using a software-defined framework for integrated space–terrestrial satellite communication was proposed with a decision factor $u_{a}$ for choosing a path based on the type of service as
$$u_{a}=w_{1}BandWidth-w_{2}Delay-w_{3}PacketLoss,$$
where $w_{1}$, $w_{2}$, and $w_{3}$ are weights [39]. A sustainable heuristic algorithm was proposed to satisfy the QoS requirements of multi-layered satellite network users to select a path based on an evaluation function fee defined as below.
fee=αdelay+βutilization+γpacketLossRate,
where α, β, and γ are balance weights [40]. A software-defined space–ground-integration-network-based deep reinforcement learning algorithm was proposed, which computes reward *r* to evaluate the performance of a link as
$$r=\alpha \;bandwidth+\beta \;throughput+\gamma \;\frac{1}{delay}+\delta \;\frac{1}{packetLossRate}+\epsilon \;\frac{1}{jitter},$$
where α, β, γ, δ, and ϵ are adjustment factors [41]. A priority-and-failure-probability-based (PFPR) routing was proposed to fulfill the QoS requirements of different services in LEO/MEO satellite communication links using an objective function penalty() that uses a linear sum as below.
$$min:penalty\left(\right)=\sum service_{j}\;Fault_{busines(j)}\;+\gamma \sum \delta \left(l\right)C\left(l\right),$$
where $service_{j}$ is the weight of different priority services, $Fault_{business(j)}$ is the failure rate, gamma is the weight for the delay, δ(l) denotes the presence or absence of a link *l*, *j* is the service type, and C(l) is the delay on the link [42]. A centralized QoS-aware algorithm using the software-defined networking controller was proposed with a linear cost function:$$score=k_{1}\;AB+\frac{k_{2}}{latency}+k_{3}\;\left(1-PDR\right)+\frac{k_{4}}{jitter}+k_{5}\;stability--flag,$$
where AB is the available bandwidth of a link, PDR is the packet loss rate, stability-flag denotes the presence of an intra-orbit link, and k1, k2, k3, and k4 are the importance weights for each metric [43]. A multi-QoS adaptive routing algorithm was proposed using an SDN-based satellite networking architecture to satisfy the QoS requirements of different types of services. A link’s weight is calculated using a linear combination as
W=αdelay+βlogloss,
where alpha or β will be increased to improve their proportion if the delay or loss rate cannot be satisfied by the flow demand in the previous path [44]. An agent-based load balancing and QoS routing for LEO satellite networks was proposed to use a cost function:$$C_{i}=\lambda \;\frac{TotalDelay(p_{i})}{\sum_{i=1}^{k}TotalDelay\left(p_{i}\right)}+\mu \;\frac{TotallLoss(p_{i})}{\sum_{i=1}^{k}TotalLoss\left(p_{i}\right)},$$
where λ and mu are weights and *i* is the link in a path *p* [45]. A routing model based on the membership function of uncertain links in mobile edge computing satellites networks has been proposed and solved using a grey wolf optimization algorithm to select a link with the minimum comprehensive value of the metrics delay, packet loss rate, and bandwidth. The comprehensive evaluation of a link is given as
$$ld=\sqrt{{\left(mf_{gk}^{td}-1\right)}^{2}+{\left(mf_{gk}^{lr}-1\right)}^{2}+{\left(mf_{gk}^{bd}-1\right)}^{2}},$$
where td, lr, and bd are the membership degree functions of the delay, packet loss rate, and bandwidth [46]. The shortest path routing approach was proposed to service delay-sensitive traffic and other traffic separately [47]. This work used a linear cost function to compute the delay on a link using the queuing delay, average delay, and service time, where the average delay is computed for a link between two nodes *i* and *j* as
$$D_{avg}\;=\;\frac{D}{1-packetlossrate},$$
and
$$D_{ij}=W_{q1_{j}}+D_{avg_{ij}}+\frac{1}{\mu },$$
where $W_{q1_{j}}$ is the queuing delay and μ is the service time.

We noticed that most of the approaches were simulation based, whereas we performed an experimental-based study with physical links emulated with delay, rate limiting, and packet loss.

## 3. Routing Approach Used-Snnr

### 3.1. Latency-Packet-Loss-Optimized Routing Objective

The main goal of our work is to optimize both the delay and packet loss in delivering satellite data using multiple paths comprised of the ISLs and IOLs between the satellite node, *s*, and a ground station, *g*. Between *s* and *g*, multiple satellite gateway nodes are connected by ISLs or IOLs to relay or route data bundles via multiple paths. The selection of a link/path impacts the end-to-end response time and the packet loss percentage of delivering data depending on the atmospheric condition on the links, the distance between two gateway nodes, the processing time by the gateway nodes, the traffic on the links, contact availability, and the sending rate. In the CSG approach, the routing algorithm runs on each gateway node and selects an outbound link based on the routing goal. From a gateway node *u* to another gateway node *v* along a path from *s* to *g*, selecting a link *i* from a set of links i=1,2,…,N at an instance *t* involves a cost, $cost_{i}^{t}$ due to the delay and the packet loss at that instance. The cost of using a link *i* at a time *t*, $cost_{i}^{t}$, is computed as follows. The delay on the link *i* to deliver a bundle from *u* to *v* at time *t* is computed as
(1)$$delay_{i}=transmissionTime_{i}^{t}+stallTime_{i}^{t},$$
where $transmissionTime_{i}^{t}$ includes the queuing delay at the egress at node *u* and $stallTime_{i}^{t}$ is the disruption time associated with the link *i* based on the contact plan using the shortest path from *s* to *g* at time *t*. The packet loss ratio is computed as
(2)$$bundleLossRatio_{i}^{t}=\frac{\left(forwardBundlesSent_{i}^{t}-backwardReportsReceived_{i}^{t}\right)}{forwardBundlesSent_{i}^{t}},$$
where $forwardBundlesSent_{i}^{t}$ represents the number of data bundles sent on the link *i* from *u* to *v* at *t* and $backwardReportsReceived_{i}^{t}$ represents the acknowledgment reports that specify the number of bundles received by *v* on the link *i* at *t*.

Our goal is to select a link *i* with a minimum cost for delivering data bundles from *u* to *v* from the set of available multiple links i=1,2,…,N at each instance *t*. In this work, we computed the $cost_{i}^{t}$ of delivering each bundle using the simplified inverse Kleinrock’s power metric proposed in [20]. With the routing approach of snnr, the $cost_{i}^{t}$ of sending a bundle on a link *i* at an instance *t* is computed using the ratio of (Equation 1) and (Equation 2) as below.
(3)$$cost_{i}^{t}=\frac{delay_{i}^{t}}{\left(1-bundleLossRatio_{i}^{t}\right)},$$
where the denominator $\left(1-bundleLossRatio_{i}^{t}\right)$ represents the number of bundles successfully delivered on the link at *t*. If the $bundleLossRatio_{i}^{t}$ on link *i* is high, the denominator will be a small value, leading to an increased delay. In contrast, if the $bundleLossRatio_{i}^{t}$ is small, the denominator will be a high value, leading to less delay. Thus, the denominator of the equation (Equation 3) helps achieve the combined optimization of both the delay and packet loss. The goal is to select a link *i* at each time instance *t*, i.e.,
(4)$$\min_{1\leq i\leq N}cost_{i}^{t}$$

### 3.2. System Implementation

The CSG approach uses a spiking neural network and reinforcement learning to select a path that delivers each bundle with optimized delay and packet loss. An SNN agent runs on each satellite, monitors the network performance on the links, and selects a link autonomously to deliver each bundle.

The cognizant agent observes the environment by computing the latency and the packet loss ratio after forwarding a bundle on an outgoing link. It estimates the cost of delivering the bundle using the metrics according to the optimization goal. The agent compares the current cost with the moving average value of the cost up to the previous instance and rewards the link if the current cost is less; else, it penalizes the link. The reward/penalty is used to update the weights of a neuron’s synapse, and the neuron that emits a spike at the earliest is used to select an outbound link next time. Thus, the performance metrics, delay, and packet loss ratio associated with each link are encoded in the time of emitting spikes by the neurons. The cognizant agent learns to choose a neuron corresponding to the outbound link, which gives the minimum latency and packet loss through repeated interactions with the environment. The interaction of the cognizant agent with its environment is depicted in Figure 1. In our context, the environment is the space communication network, which includes spacecraft and relay satellites on different orbits, surface elements on planets such as Mars and the Moon, and ground stations on the Earth, as shown in the right side of the picture.

Each node computes the transmission time of a bundle’s delivery on an incoming (ingress) link when it receives a forward report by calculating the difference between the time stamp of the previous node, which sends the report, and the time at which it received the report. This transmission time is sent back in a backward report to the previous node from which it received the forward report on the same link. The SNN agent counts the number of forward reports sent and the number of backward reports received on each outbound link and computes the packet loss ratio on the link. We compared the performance of snnr routing with the routing approaches that use the following cost functions: 1. snndp: a linear sum (Equation 5); 2. snnd: delay-only (Equation 6); 3. snnp: packet-loss-only (Equation 7); 4. static: CGR;

snndp—optimizes both the response time and packet loss using a linear cost function as below:
(5)$$cost_{dp_{i}}^{t}=k\;bundleLossRatio_{i}^{t}+\left(1-bundleLossRatio_{i}^{t}\right)\left(transmissionTime_{i}^{t}+stallTime_{i}^{t}\right),$$
where *k* is a constant used to balance the two metrics, i.e., bundleLossRatio is a decimal value, whereas $transmissionTime_{i}^{t}$ is measured in milliseconds;snnd—optimizes only the response time in delivering data from a satellite node to a ground station. The optimization of $cost_{d}$ to optimize the delay is computed for an outbound link *i* at a satellite node at *t*, computed as
(6)$$cost_{d_{i}}^{t}=transmissionTime_{i}^{t}+stallTime_{i}^{t};$$snnp—optimizes only the packet loss in delivering data. The optimization $cost_{p_{i}}^{t}$ to minimize the packet loss ratio is computed for an outbound link *i* at a satellite node at *t* as
(7)$$\begin{array}{cc} bundleLossRatio_{i}^{t} & =\frac{\left(forwardBundlesSent_{i}^{t}-backwardReportsReceived_{i}^{t}\right)}{forwardBundlesSent_{i}^{t}}\\ cost_{p_{i}}^{t} & =bundleLossRatio_{i}^{t}; \end{array}$$static—this is the same as CGR, which selects the earliest link in the shortest path between two satellite nodes.

## 4. Experimental Methodology

The details of the satellite nodes participating in a communication mission and the topology details are described in a JSON file named contactPlan.json. The JSON file contains information such as the source, destination, availability of the satellite contacts (line-of-sight) to each other, data rate, start time, and end time of the contacts. This file will be distributed to all nodes from a mission-specific control center through a separate channel and updated at scheduled intervals [48]. The SNN on each node decides on an outbound link from the set of available links to the next hop on the shortest path from the source node to the destination node, computed using the contacts’ availability from the contact plan file. The SNN autonomously learns to select outbound links at each node based on their performances in the previous instances.

### Experimental Setup

We emulated an Earth observation data collection scenario in our lab testbed using virtual machines connected in two topological configurations: 1. Topology 2. These are shown in Figure 2. The satellite h23 is collecting Earth observation data from space and sending them to a ground station (GS) h25 on the Earth.

We used the Linux traffic control utility NetEm to emulate longer delays, channel attenuation, and the bandwidth characteristics of the space network by configuring the delay, rate limiting, and packet loss for each test scenario differently. The network topology represents the multiple paths between satellites using ISLs and IOLs. We used the Bpgen tool to generate bundles of size 10,000 bytes and sent at rates from 1 bundle/s to 10 bundle/s, and each experiment was run for 180 s and repeated six times to have a statistical average of the network performance metrics. The sending rate implies the number of concurrent users of the system. We used the UDP protocol, since it does not support re-transmissions for lost bundles, for which we needed to calculate the packet losses strictly. We present the performance of the SNN algorithms with different optimization goals under different scenarios in the Results Section.

## 5. Results

### 5.1. Topology 1: Results

The satellite node h23 is connected to the ground station node h25 using ISLs and satellite gateways h26, h29, h30, and h27, as shown in Figure 2a. The configured delay, rate limiting, and packet loss under different test scenarios are depicted in Table 1.

#### 5.1.1. Scenario 1

In Scenario 1, a delay of 100 ms was configured on each link h26–>h30–>h27, and a delay of 50 ms was configured from h26–>h29–>h27. The static method was configured to select the path h26–>h29–>h27.

Average response times obtained from all routing methods were identical except at bundle rate 10 bundles/s, as depicted in Figure 3a. Since the sending rate was within the bandwidth capacity of the links, the average response time looks identical, even though the SNN algorithms selected 10% of the time h30 for exploring globally optimal links. Since there was no packet losses configured, the average packet loss percentage was also negligible and similar for all the routing methods, except at 10 bundles/s, as depicted in Figure 3b. Throughput was also similar for all the routing methods in this scenario, as in Figure 3c.

Figure 4 depicts the link selection at Node h26 by the SNN algorithms according to their optimization goals in Scenario 1. All SNN algorithms selected h29 more times than h30 since the delay was better on the link connecting from h26 to h29 when compared to h30.

#### 5.1.2. Scenario 2

In Scenario 2, the average response times achieved with the routing methods are depicted in Figure 5a. The average response time with snnr was 9–20% better than snndp at lower bundle rates, but 13–27% worse at bundle rates of 9–10 bundles/s. snnr and snndp showed 2–44% and 2–27% worse performance than snnd. The static and snnp methods performed better at lower bundle rates, but snndp and snnr performed 50% better than static and 25% better than snnp at 10 bundles/s.

The bundle loss percentage in shown in Figure 5b. The snnr method showed 25–43% and 13–48% better packet loss performance than snndp and snnd at bundle rates of 8–10 bundles/s. snnr and snndp showed 20–60% and 15–60% better performance than static. snnp showed 270–650% superior performance over snnr and snndp.

Figure 5c shows that the throughput with snnr was up to 24% better than snndp and snnd at bundle rates of 8–10 bundles/s. snnr showed up to 63% better throughput than static, and snndp showed up to 50% better throughput than static. snnp showed up to 38% better throughput than snnr and snndp.

Figure 6 depicts the link selection at the node h26 by the SNN algorithms according to their optimization goals in Scenario 2, and snnp selected h30 the most to avoid packet loss.

#### 5.1.3. Scenario 3

In Scenario 3, we used rate limiting in the left path from h26–>h30–>h27 to slow down the traffic on the left path, and the right path was configured with less delay and packet loss on the links h26–>h29–>h27.

Figure 7a shows that snnr performed worse than snndp and snnd at lower and higher bundle rates, but performed 5–19% and 2–7% better than snndp and snnd at bundle rates of 3–7 bundles/s. snnr showed inferior performance to snnp at lower bundles rates, but performed 36–68% better at 7–10 bundles/s. The performances of snndp and snnr were 1–47% and 23–54% better than static at bundle rates of 9–10 bundles/s, whereas static was better than both methods at lower bundle rates. snndp was 5–20% better than snnd at 9–10 bundles/s and 43–70% better than snnp at 7–10 bundles/s.

Figure 7b shows the packet loss percentage with snnr being 1–38% better than snndp and snn, and 10–50% better than static. snndp was 7–32% better than static. snnp showed 11–440% and 16–758% better performance than snnr and snndp.

Figure 7c shows the throughput with snnr being up to 25% better than snndp and snnd and 10–50% better than static. Throughput with snnp was 7–40% better than snnr and snndp. snndp showed 7–30% better throughput than static.

Figure 8 depicts the link utilization at Node h26 in Scenario 3. We can see that snnp selected h30, the most to avoid packet loss on the link connecting h29, whereas other SNN algorithms selected h29 more since their optimization goal includes delay.

#### 5.1.4. Scenario 4

In Scenario 4, the links on the right path h26–>29–>h27 were emulated with less delay and no packet loss, whereas the left path h26–>h30–>h27 had more delay including rate limiting.

Figure 9a shows that the average response time with snnr was slightly inferior to snndp and snnd. snndp performed 1–11% worse than snnd and 1–20% better than snnp at 1–8 bundles/s. snndp and snnr showed 11–24% and 4–24% better performance than static at bundle rates of 9–10 bundles/s.

Figure 9b shows the bundle loss percent was the same for all methods for bundle rates up to 8 bundles/s. Figure 9c shows that all routing methods showed similar throughput at all bundle rates.

Figure 10 depicts the link selection at the node h26 by the SNN algorithms according to their optimization goals in Scenario 4.

#### 5.1.5. Scenario 5

In Scenario 5, the emulated delay on the right path h26–>h29–>27 was higher than the left path h26–>h30–>27, but the left path was rate limited by configuring queues on the output port of h26 and h30. The static method was configured to select the left path h26–>h30–>h27.

Figure 11a shows the average response time with snnr being 4–65% better than static. snnr showed 2–23% worse performance than snndp and snnp. snnr was 1–12% worse than snnd. snndp performed 6–60% better than static and 4–19% worse than snnd and snnp.

Figure 11b shows the bundle loss with static being up to 100% higher than snnr and snndp at bundle rates of 8–10 bundles/s. Both snnr and snndp showed the same packet loss percentage at lower bundle rates, but snnr showed 23–100% better performance than snndp.

Figure 11c shows that all routing methods showed similar throughput at all bundle rates except at higher bundle rates, where all the SNN methods showed 20% better performance than static.

Even though the delay from h26 to h30 was less than h29, the configured rate limiting slowed down the bundle delivery to h30; hence, snnd, snndp, and snnr selected h29 most of the time, whereas snnp selected h29 and h30 equally, as shown in Figure 12.

#### 5.1.6. Scenario 6

In Scenario 6, the links on the path h26–>h30–>27 were emulated with less delay and rate limit and a packet loss of 10%, whereas the links on the right path h26–>h29–>27 were configured with a long delay and no packet loss. static was configured to select the h26–>h30–>27 path.

Figure 13a shows that the average response time with snnr was 15–40% and 2–35% better than snndp and snnd, except at the bundle rate of 10 bundles/s. Both snnr and snndp showed worse performance than static at lower bundle rates and up to 80% better performances for bundle rates of 8–10 bundles/s. snndp was worse than snnd at lower bundle rates, but showed 7–41% better response at 9–10 bundles/s. snndp was 28–100% worse than snnp.

Figure 13b shows the packet loss percentage with snnr being up to 50–93% better than snndp and snnd. snnr showed 1–130% worse performance than snnp, but showed 20–66% better performance at bundle rates of 1–3 bundles/s. snnr and snndp showed 65–95% and 14–60% better packet loss percentage than static. snndp showed up to 50% and 500% inferior performance to snnd and snnp.

Figure 13c shows that snnr had 30–200% and 15–130% better throughput than snndp and snnd. snnp had 2–17% and 23–60% better throughput than snnr and snndp. snndp showed up to 47% lower throughput than snnd. Both snnr and snndp showed 143–381% and 17–267% better throughput than static.

Figure 14 shows that snnd and snndp selected h30 most since both intended to optimize delay, whereas snnr and snnp selected h29 most.

### 5.2. Topology 2

In Topology 2, h29 is connected to h30 using a bandwidth capacity of 10 Mbs without any configured delay, as depicted in Figure 2b. The delay and packet loss configurations are provided in Table 2.

#### 5.2.1. Scenario 7

In this test scenario, each link on the path h26–>h30–>h27 was configured with a delay of 100 ms, and h26–>h29–>h27 was configured with a delay of 50 ms with a 10% packet loss. The link between h29 to h30 had no delay or packet loss. The static method was configured to use the path h26->h29->h27.

Figure 15a shows the average response time with snnr being 1–30% better than snndp and 2–25% better than snnd. Both snnr and snndp performed worse than static at lower bundles, but showed 36–70% and 36–67% better response for bundle rates of 9–10 bundles/s. snndp showed up to 20% better performance than snnd. snndp and snnr showed up to 100% and 90% worse performance than snnp.

Figure 15b shows the packet loss percentage with snnr being 2–46% and 2–25% better than snndp and snnd. snndp had 8% better performance than snnd. snnp showed up to 100% better packet loss performance than snnr and snndp. snnr and snndp showed 41–70% 29–43% better performance than static.

Figure 15c depicts that the throughput with snnr being up to 4–31% and 2–27% better than snndp and snnd. snnp showed up to 30% better throughput than snnr and snndp. snndp showed up to 10% better throughput than snnd. snnr and snndp showed 105–176% and 92–144% better throughput than static.

#### 5.2.2. Scenario 8

In this scenario, the delay and packet loss were the same as in Scenario 7, but the connection between h29 and h27 had interruptions every other 15 seconds. Hence, we expected the SNN algorithms that selected h29 from h26 to prefer h30 at h29 whenever there was a connection interruption. static was configured to use the left path h26–>h29–>h27.

The average response times achieved with the routing methods in this scenario are depicted in Figure 16a. We can see that the average response times increased with all the routing methods compared to Scenario 7 due to the connection interruptions between h29–>h27. snnr showed 12–69%, 8–52%, and 17–80% better average response times than snndp, snnd, and snnp. snnr and snndp showed 36–78% and 20–65% better performance than static. snndp showed 8–67% better performance than snnp, but performed worse than snnd.

Figure 16b shows that the packet loss percentage achieved by snnr was 2–76% and 1–125% worse than snndp and snnd. snnp showed 2–140% better performance than snnr for bundle rates of 1–8 bundles/s, whereas snnr was 18–25% better than snnp at bundle rates of 8–10 bundles/s. Both snnr and snndp were 45% better than static. snndp showed 12–14% better performance than snnd at 1–2 bundles/s, but performed worse at the higher bundle rates, whereas it performed worse at the lower bundle rates than snnp and 20% better at 8–10 bundles/s.

The packet loss percentage was better than in the previous scenario due to the duplicate transmissions during connection interruptions. The DTN caused the bundles to be stored during connection interruptions and forwarded when the connection was restored.

Figure 16c shows that the throughput with snnr was 1–25% worse than snndp and snnd. snnp showed up to 33% better throughput at bundle rates of 1-8 bundles/s, and snnr showed 12–21% better throughput at higher bundle rates. snndp showed 1–20% better throughput than static, and snnr showed 10–18% better throughput only at 9–10 bundles/s. snndp showed 2–23% better throughput than snnp at 7–10 bundles/s.

All SNN methods showed better average response times and packet-loss percentages than the static method because of utilizing the alternate links during connection interruptions.

#### 5.2.3. Scenario 9

In this scenario, both paths were alternatively configured with low delay, rate limiting, and packet loss on one of the links and higher delay on the other link, as depicted in Table 2. The static method was configured to select the h26–>h30–>h27 path, whereas the SNN algorithms made dynamic selections at h26, h29, and h30 to choose an outbound link.

Figure 17a shows that snnr was 9–30% better than snndp, but 153% worse at 10 bundles/s. snnr was 10–20% better than snnd, except at 10 bundles/s, where it was 78% worse. snnr was 3–126% worse than snnp. The average response time with snnr was 2–79% better than static, whereas snndp was 47–80% better at 8–10 bundles/s. snndp showed 1–29% better performance than snnd and 14–59% worse performance than snnp.

Figure 17b shows the bundle loss percentage with snnr being 33–80% and 13–81% better than snndp and snnd. snnp achieved the best performance by reducing 210% and 491% packet losses than snnr and snndp since it avoided the links with packet losses at h26 and h29. The snnr method showed 40–85% better performance than static, whereas snndp showed 8–59% better packet loss performance than static. snndp performed 4–94% worse than snnd.

Figure 17c shows the throughput with snnr being up to 17–149% and 2–105% better than snndp and snnd. snnp showed 1–25% and 15–61% better throughput than snnr and snndp. snnr showed 38–127% better throughput than static, whereas snndp showed 10–93% better throughput. snndp showed 7–18% inferior throughput to snnd.

Figure 18a shows that the path length with the SNN methods increased due to selecting the better-quality link h29->h30 through learning in scenario 7. Hence, the packet loss percentage was less with them with an increased average path length. Figure 18b shows the path lengths were higher with the SNN methods because of using the h29–>h30 link in scenario 8.

We can verify that snnp showed the maximum path lengths in Figure 18c by choosing h26–>h29–>h30–>h27 or h26–>h30–>h29–>h27 in scenario 9.

## 6. Discussion

In general, all SNN methods made 10% random selections for exploring the globally optimal choices as part of the learning and selected unfavorable links randomly. The static method was configured to choose the earliest available link from the shortest path from h23 to h25 in all test scenarios. In snndp, we used k=100, which gave us better results. We found it challenging to select a value for *k* to balance both metrics.

In Scenario 1, the beneficial path was h26–>h29–>h27. All routing methods selected that path and performed equally. At higher bundle rates, the SNN algorithms preferred h30 to load balance, which led to better performance than the static method.

In Scenario 2, a small percentage of packet loss at lower bundle rates on the less delay path made snnr select h29 and led to a slightly increased average response time than snnd. However, increased packet losses at higher bundle rates caused snnr to choose h30 more at the expense of increased delay. Since static used only h29, the increased packet loss decreased the throughput more than the SNN methods. Since snnp tried to optimize only packet loss, it selected h30 and performed better.

In Scenario 3, configured rate limiting on the link to h30 slowed down the bundle delivery and raised the delay more than on the link to h27. Hence, by selecting h30 to avoid bundle losses while trying to reduce delay, snnr, snndp, and snnd performed worse than snnp, but better than static. snnp achieved better packet loss and throughput with increased average response time by selecting h30 more.

Since the network conditions favored h29 as the profitable link, all routing methods selected the path to h29 in Scenario 4. All SNN methods sent some of the bundles to h30 at random for exploration and showed better performance than static at higher bundle rates by load sharing between the two links.

In Scenario 5, the link to h30 delayed the bundle delivery and caused a higher delay. The snnr method selected h29 more times, similar to the other SNN methods, and showed similar performance at lower bundle rates. However, at higher bundle rates, snnr used h30 more than the other SNN methods and improved the throughput by increasing the average response time. All SNN methods showed better performance than static by using h30 randomly.

In Scenario 6, though the delay on the path h26–>h29–>h27 was high, there was no packet loss, which caused the denominator in the snnr cost function to be high and caused it to pick h29 more. Hence, snnr resulted in less average response time, packet loss percentage, and high throughput.

In Scenario 7, snnr showed slightly better performance than snndp and snnd. All SNN methods showed better packet loss and throughput performance than static.

In Scenario 8, all SNN methods showed better average response times and packet loss percentages than static. The static method achieved the same throughput as the SNN methods, though it was configured to select the path with losses. Interrupted connections between h29 and h27 caused bundles to be stored and delivered later by the DTN protocol and caused the throughput to be the same. At lower bundle rates, snnr, snndp, and snnd preferred h29 over h26 and selected h27 or h30 according to the connection interruptions at h29. At higher bundle rates, snnr, snndp, and snnd sent more bundles to h30 from h26, then snndp and snnd selected h30–>h29–>h27, but snnr selected h30–>h27. Thus, snnr achieved less average response time than snndp and snnd. snnp achieved better throughput at lower bundle rates and decreased throughput at higher bundle rates by choosing the path h26–>h30–>h29–> most.

In Scenario 9, snnr and snnp preferred the h26–>h29–>h30–>h27 path at all bundle rates, whereas snnd and snndp preferred h26–>h30–>h27 at lower bundle rates and h26–>h29–>h30–>h27 at higher bundle rates. Hence, snndp and snnd resulted in high packet losses and lower throughput, whereas snnr and snnp achieved better performances.

snnp achieved the minimum packet loss in all scenarios according to its optimization goal, but with increased response times in Scenarios 3 and 8. snnd had only latency as its optimization goal and showed better response times in Scenarios 3, 6, and 8. The response time plots did not show much difference in general because the induced delays on the links did not cause much difference in the response times for the sending rates and the available capacities. The snndp and snnr algorithms had both delay and packet loss as their optimization goals and tried to optimize both metrics concurrently through the desired balancing.

We can see that snnr performed equally to the other SNN algorithms when the link qualities were uniform and performed better when the links on the path had conditions for which is was challenging to make decisions, such as in Scenario 6 and Scenario 9.

## 7. Conclusions

Our experimental results proved that, by using inverse Kleinrock’s power metric as a cost function in an SNN with reinforcement learning, it is possible to achieve concurrent optimization of both delay and packet loss percentage in a satellite network with uncertain conditions. The CSG approach helps to make forwarding decisions autonomously adaptive to the changing environment to deliver data with a better QoS.

Our future work will consider exploring offline learning predictions whenever online feedback paths are longer, as in deep space communications such as between Earth and Mars.

## Figures and Tables

**Figure 1 sensors-23-00310-f001:**
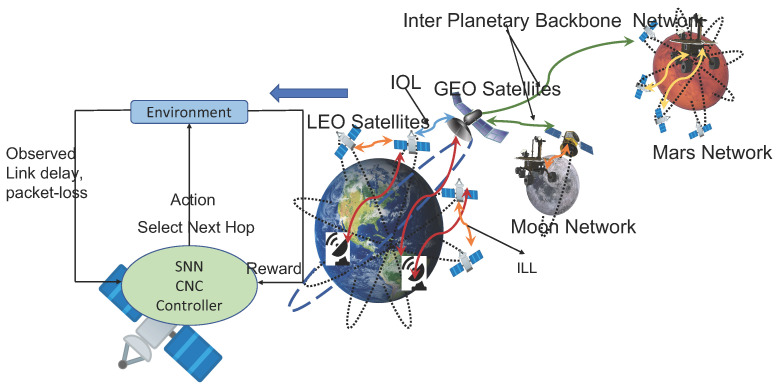
Cognizant controller running on satellites autonomously selects links at each satellite to optimize latency and pack loss in a possible interplanetary network.

**Figure 2 sensors-23-00310-f002:**
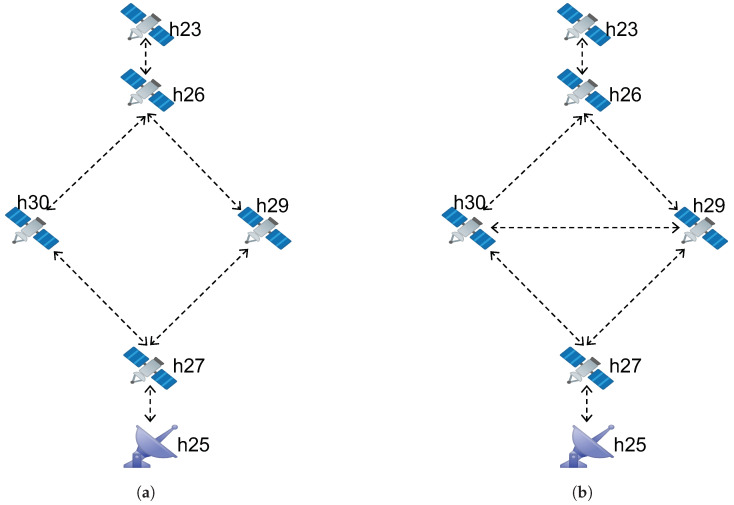
Space-based information network of satellites and a GS. (**a**) Topology 1 (**b**) Topology 2.

**Figure 3 sensors-23-00310-f003:**
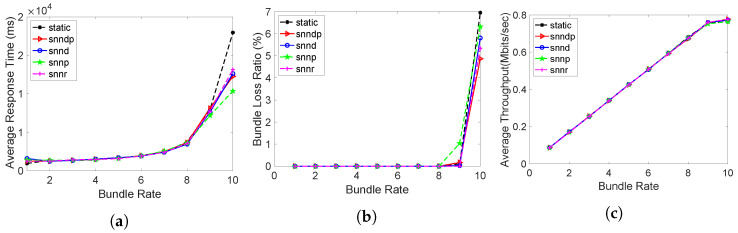
Routing performance: Scenario 1. (**a**) Average response time; (**b**) Packet loss ratio (%); (**c**) Throughput.

**Figure 4 sensors-23-00310-f004:**
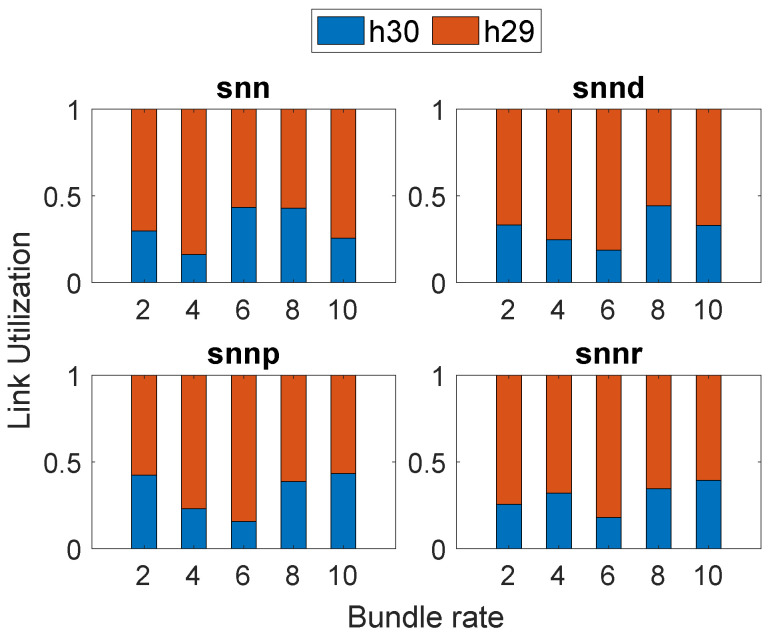
Link selection at Node h26: Scenario 1.

**Figure 5 sensors-23-00310-f005:**
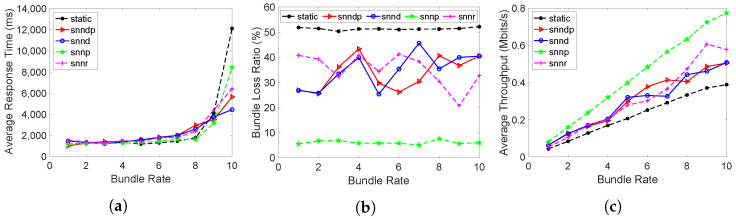
Routing performance: Scenario 2. (**a**) Average response time; (**b**) Packet loss ratio (%); (**c**) Throughput.

**Figure 6 sensors-23-00310-f006:**
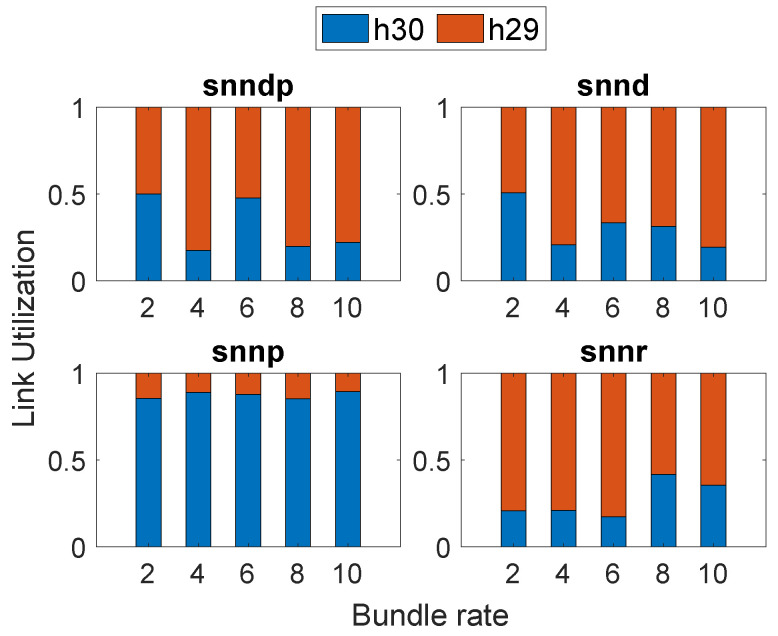
Link selection at Node h26: Scenario 2.

**Figure 7 sensors-23-00310-f007:**
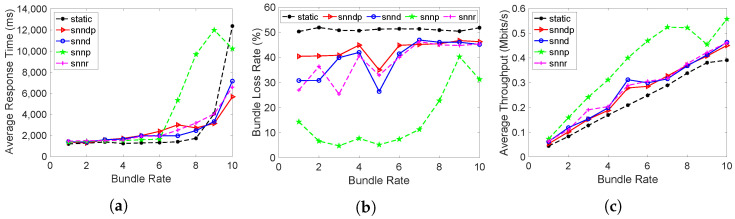
Routing performance: Scenario 3. (**a**) Average response time; (**b**) Packet loss ratio (%); (**c**) Throughput.

**Figure 8 sensors-23-00310-f008:**
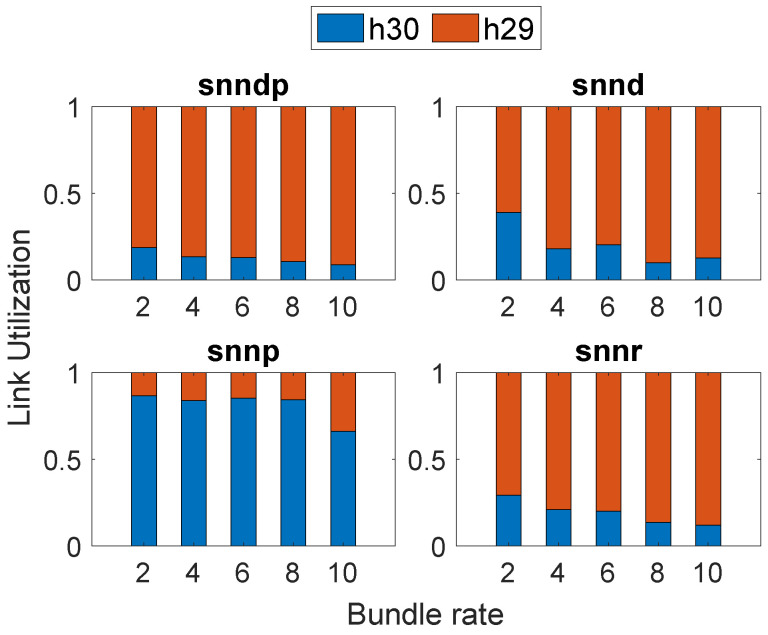
Link selection at Node h26: Scenario 3.

**Figure 9 sensors-23-00310-f009:**
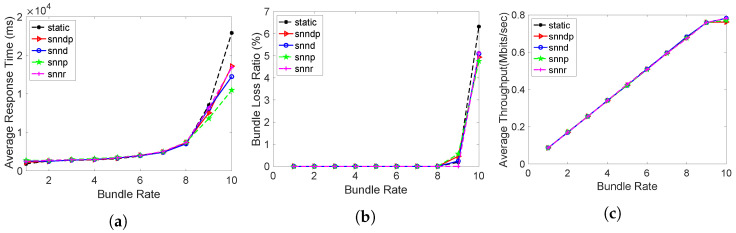
Routing performance: Scenario 4. (**a**) Average response time; (**b**) Packet loss ratio (%); (**c**) Throughput.

**Figure 10 sensors-23-00310-f010:**
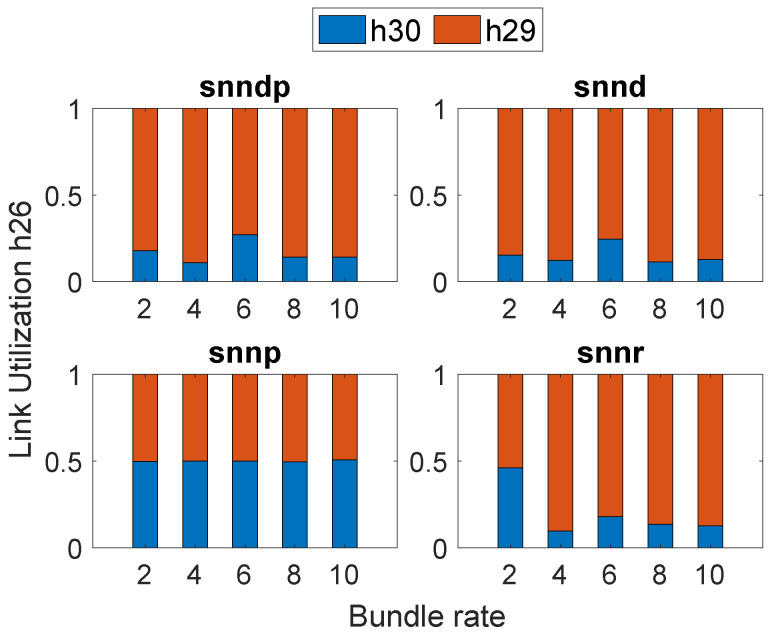
Link selection at Node h26: Scenario 4.

**Figure 11 sensors-23-00310-f011:**
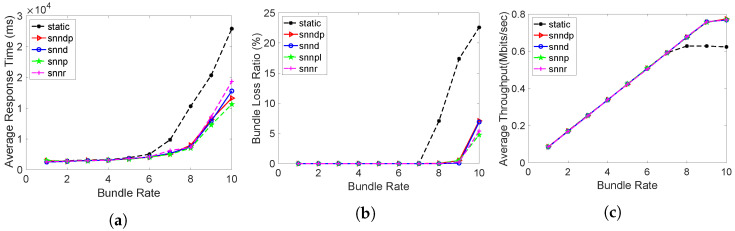
Routing performance: Scenario 5. (**a**) Average response time; (**b**) Packet loss ratio (%); (**c**) Throughput.

**Figure 12 sensors-23-00310-f012:**
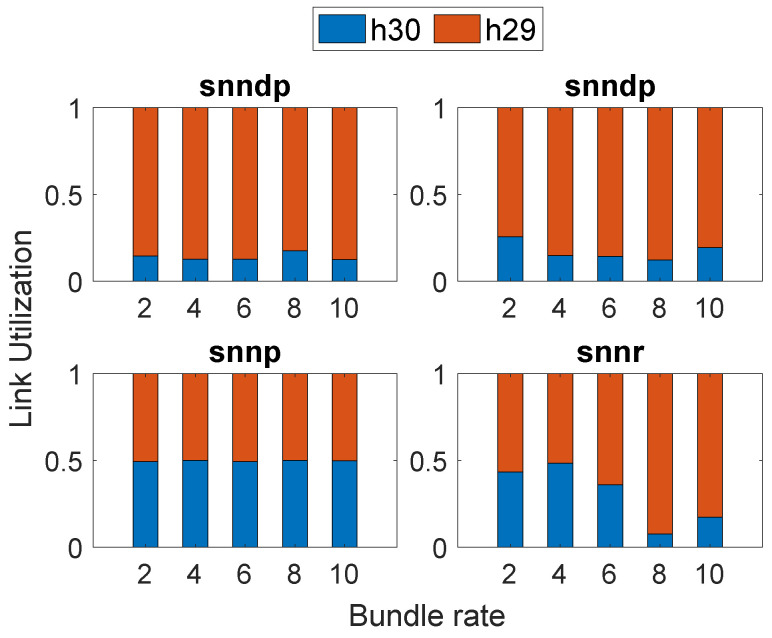
Link selection at Node h26: Scenario 5.

**Figure 13 sensors-23-00310-f013:**
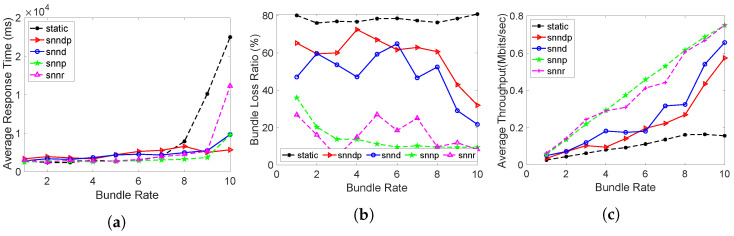
Routing performance: Scenario 6. (**a**) Average response time; (**b**) Packet loss ratio (%); (**c**) Throughput.

**Figure 14 sensors-23-00310-f014:**
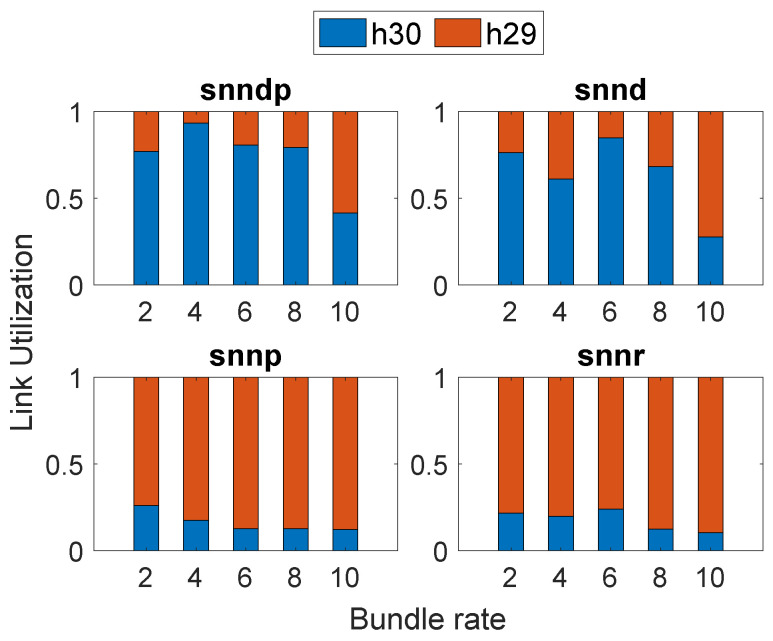
Link selection at Node h26: Scenario 6.

**Figure 15 sensors-23-00310-f015:**
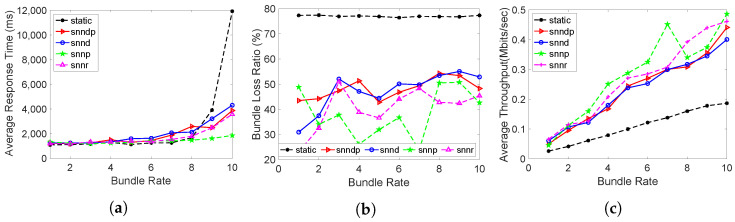
Routing performance: Scenario 7. (**a**) Average response time; (**b**) Packet loss ratio (%); (**c**) Throughput.

**Figure 16 sensors-23-00310-f016:**
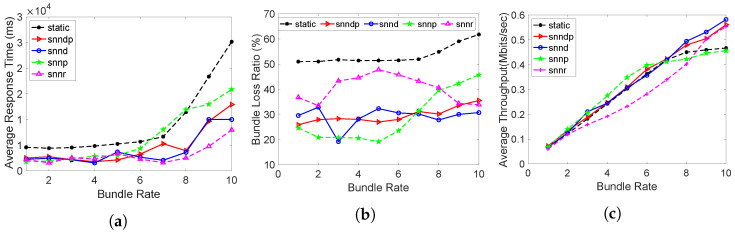
Routing performance: Scenario 8. (**a**) Average response time; (**b**) Packet loss ratio (%); (**c**) Throughput.

**Figure 17 sensors-23-00310-f017:**
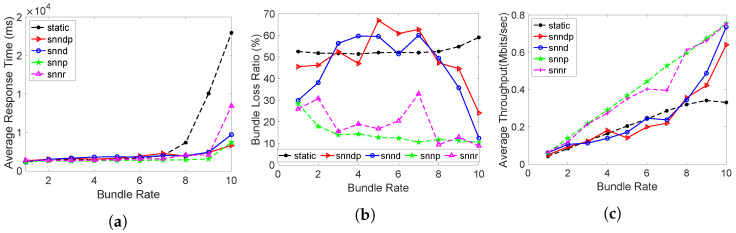
Routing performance: Scenario 9. (**a**) Average response time; (**b**) Packet loss ratio (%); (**c**) Throughput.

**Figure 18 sensors-23-00310-f018:**
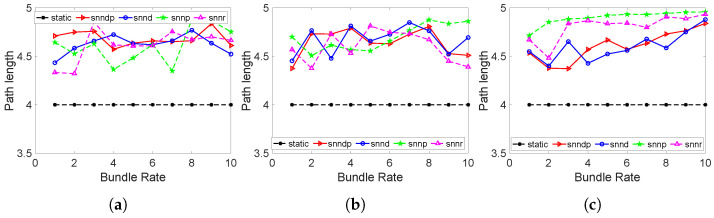
Path lengths traveled by bundles in Topology 2. (**a**) Scenario 7; (**b**) Scenario 8; (**c**) Scenario 9.

**Table 1 sensors-23-00310-t001:** Delay and packet-loss configurations tested with Topology 1.

Scenario	h26–h29	h29–h27	h26–h30	h30–h27
Scenario 1	50 ms	50 ms	100 ms	100 ms
Scenario 2	50 ms 5% loss	50 ms 5% loss	100 ms	100 ms
Scenario 3	50 ms loss 5%	50 ms loss 5%	120 ms 512 Kb rate limit	120 ms 512 Kb rate limit
Scenario 4	50 ms	50 ms	120 ms 640 Kb rate limit	120 ms 640 Kb rate limit
Scenario 5	120 ms	120 ms	50 ms 640 Kb rate limit	50 ms 640 Kb rate limit
Scenario 6	200 ms	200 ms	50 ms 640 Kb rate limit	50ms 640 Kb rate limit
			loss 10%	loss 10%

**Table 2 sensors-23-00310-t002:** Delay and packet loss configurations tested with Topology 2.

Scenario	h26–h29	h29–h27	h26–h30	h30–h27
Scenario 7	50 ms loss 10%	50 ms loss 10%	100 ms	100 ms
Scenario 8	50 ms loss 10%	50 ms loss 10%	100 ms	100 ms
Scenario 9	200 ms	50 ms loss 10%	50ms loss 10%	200 ms
		rate 640 kbit	rate 640 kbit	

## Data Availability

The data presented in this study are available on request from the corresponding author. The data are not publicly available due to privacy restrictions.

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
