# Peer review of "Delay-Packet-Loss-Optimized Distributed Routing Using Spiking Neural Network in Delay-Tolerant Networking"

_sensors, 2022, doi:10.3390/s23010310_

Round 1

Reviewer 1 Report

The authors proposed a work on routing using Spiking Neural Network in Satellite Communication. This is an interesting and important topic. The object orientation of this research is a real problem for NASA. The methods are certainly adequately described, and the results are clearly presented. I suggest that the article be considered for publication after revisions.

Here are some suggestions:

1)     The coordination of multi-layer satellite networks requires efficient cross-network cooperative management and control. Therefore, the authors need to supplement the control node distribution and architecture of routing in the system model. The influence of synergy on routing should be further explained.

2)     The model of the time-sensitive business needs further description. In the sacce information communication service scenario, the number of concurrent users should be considered, which will affect the queuing delay. I suggest that the author needs to add this model to highlight the novelty of the article.

3)     The author needs to further describe how to design SNN, the structure and configuration of the hidden layer,

4)     How to solve the constraints and objective functions after the author constructs them.

Author Response

  • The coordination of multi-layer satellite networks requires efficient cross-network cooperative management and control. Therefore, the authors need to supplement the control node distribution and architecture of routing in the system model. The influence of synergy on routing should be further explained.

Ans: The satellite nodes participating in a communication system is declared in a JSON file named contactPlan.json. The JSON file contains information such as the availability of the satellite links to each other, data rate, and start and end time of the availability of the links. This file will be sent to each node from the ground network control center through a separate connection and updated in scheduled time intervals.

Each node decides an outbound to the next hop based on the performances from the previous instances through autonomous learning on the shortest path. The shortest path is computed before each outbound selection using the connection availability from the contact plan file.

We added a paragraph about the contact plan in section 4.

  • The model of the time-sensitive business needs further description. In the sacce information communication service scenario, the number of concurrent users should be considered, which will affect the queuing delay. I suggest that the author needs to add this model to highlight the novelty of the article.

Ans: Each experiment is tested with sending rates from 1 bundle/second to 10 bundles/second, which represents the concurrent users of the system. We mentioned in the paper that transmissionTimeit includes queueing delays at the egress of the previous node.

  • The author needs to further describe how to design SNN, the structure and configuration of the hidden layer.

Ans: We directed the readers to our previous work that describes the SNN design.  We also briefly described the basics of SNN in section 1.1.

4)     How to solve the constraints and objective functions after the author constructs them.

      Ans: SNN is a self-learning-based neural network that makes adaptive decisions based on the performances of the previously selected outbound links.

Reviewer 2 Report

Summary:

The applications that benefit from satellite communication are Earth observation (EO), military missions, disaster management, and 5G/6G integration, to name a few. Some essential applications, such as Earth observation (EO), depend on time-sensitive and error-free data delivery, which needs better throughput connections. It is challenging to route space data to ground stations with better QoS by leveraging the inter-satellite links (ISL) and inter-orbital links (IOL). Routing approaches that use the shortest path to optimize latency may cause packet losses and reduced throughput based on the channel conditions, while routing methods that try to avoid packet losses may end up delivering data with long delays. Existing routing algorithms that use multi-optimization goals tend to use priority-based optimization to optimize either of the metrics. Hence, the critical satellite missions that depend on high throughput and low-latency data delivery need routing approaches that optimize both metrics concurrently.

The main goal of this paper is to optimize both the delay and packet loss in delivering satellite data using multiple paths comprised of ISL and IOL between the satellite node and a ground station. This paper used a modified version of Kleinrock’s power metric to show that it is possible to reduce both metrics, i.e. high throughput and low-latency data delivery, using experimental evaluations. This paper used a cognitive space routing approach, which uses a reinforcement learning-based spiking neural network to implement routing strategies in NASA’s High Rate Delay Tolerant Networking (HDTN) project.

The aim of the paper:

·         This paper considers packet losses due to the channel characteristics while all satellite nodes have equal buffer capacity.

·         This paper tried to optimize both latency and packet loss together without using any weights or priorities by using the simplified Kleinrock’s power metric in the snnr routing algorithm, which uses a ratio of delay and packet loss.

·         This paper compared the performance of the snnr with the other routing approaches with the following goal functions :

o   linear combination of delay and packet loss,

o   delay-only,

o   packet loss-only,

o   Contact Graph Routing (CGR) that uses the shortest path with the earliest available links.

·         This paper evaluated the performance of the routing approaches in a laboratory testbed with virtual machines under various testing scenarios with different emulated delay, packet loss, and bandwidth configurations.

·         This paper implemented a proposed routing algorithm using the CSG routing approach in HDTN software run on each virtual machine

Strengths:

1-Good choice of research subject.

2-Good proposed idea. 

3-Detailed methods, experimentations, and result explanations.

4-Excellent work.

Weaknesses:

1-The CSG should be defined and explained earlier than section 4.2, preferably in section 3,

2-snnr routing algorithm requires further clarifications and explanations.

Should the snn, snnr,…etc be written as an upper case?

3-At the discussion section, the authors did a good work of showing the results of all scenarios. It will be better to summarize all scenarios as a whole after explaining each scenario individually.  In that whole summarization, the authors should show how the results linked to the previously mentioned goals and aims of the paper.

4-At the conclusions section, the part (Our future work will consider exploring offline learning predictions whenever online feedback paths are longer, as in deep-space communications such as between Earth-Mars.) should be written as a separate paragraph.

5-Can you support us with the URL of your testbed and a place on GitHub, for example, that contains data, results, and other materials used in your research?

Author Response

1-The CSG should be defined and explained earlier than section 4.2, preferably in section 3,

Ans: We moved the CSG details to section 1 since we want section 3 to describe only the system implementation.

2-snnr routing algorithm requires further clarifications and explanations.

Ans: The explanation was already there in the paper but without a title. We added a title to the explanation of snnr in section 3.

Should the snn, snnr,…etc be written an upper case?

Ans: The spiking neural network is referred to as SNN, whereas the SNN-based algorithms with specific optimization goals are referred to as snnd, snndp, and snnr.

3-At the discussion section, the authors did a good work of showing the results of all scenarios. It will be better to summarize all scenarios as a whole after explaining each scenario individually.  In that whole summarization, the authors should show how the results linked to the previously mentioned goals and aims of the paper.

Ans: Yes, we added a paragraph in the discussion section.

4-At the conclusions section, the part (Our future work will consider exploring offline learning predictions whenever online feedback paths are longer, as in deep-space communications such as between Earth-Mars.) should be written as a separate paragraph.

Yes, we wrote the future work as a separate paragraph.

5-Can you support us with the URL of your testbed and a place on GitHub, for example, that contains data, results, and other materials used in your research?

Ans: We are in the process of porting CSG into the latest version of HDTN and the current work may be available on GitHub once the porting is completed.

Round 2

Reviewer 1 Report

I think the author has revised all the problems mentioned. Articles may be considered for acceptance.